# Segmentation of Leaves and Fruits of Tomato Plants by Color Dominance

**Juan Pablo Guerra Ibarra** [1] , **Francisco Javier Cuevas de la Rosa** [1,*] **and Oziel Arellano Arzola** [2]

[1]   Centro de Investigaciones en Óptica A.C., Leon 37150, Guanajuato, Mexico; juangi@cio.mx
[2]   Instituto Tecnológico Nacional de México, Campus Zamora, Zamora 59720, Michoacan, Mexico; oziel.aa@zamora.tecnm.mx
*   Correspondence: fjcuevas@cio.mx

**Abstract:** The production of food generated by agriculture has been essential for civilizations throughout time. Tillage of fields has been supported by great technological advances in several areas of knowledge, which have increased the amount of food produced at lower costs. The use of technology applied to modern agriculture has generated a research area called precision agriculture, which has providing crops with resources in an exact amount at a precise moment as one of its most relevant objectives The data analysis process in precision agriculture systems begins with the filtering of the information available, which can come from sources such as images, videos, and spreadsheets. When the information source is digital images, the process is known as segmentation, which consists of assigning a category or label to each pixel of the analyzed image. In recent years, different algorithms of segmentation have been developed that make use of different pixel characteristics, such as color, texture, neighborhood, and superpixels. In this paper, a method to segment images of leaves and fruits of tomato plants is presented, which is carried out in two stages. The first stage is based on the dominance of one of the color channels over the other two, using the *RGB* color model. In the case of the segmentation of the leaves, the green channel dominance is used, whereas the dominance of red channel is used for the fruits. In the second stage, the false positives generated during the previous stage are eliminated by using thresholds calculated for each pixel that meets the condition of the first stage. The results are measured by applying performance metrics: *Accuracy*, *Precision*, *Recall*, *F1-Score*, and *Intersection over Union*. The results for segmentation of the fruit and leaves of the tomato plants with the highest metrics is *Accuracy* with 98.34% for fruits and *Recall* with 95.08% for leaves.

**Keywords:** computer vision; color dominance; RGB; precision agriculture; artificial intelligence





## 1. Introduction

Agriculture is a fundamental economic activity for the subsistence of human beings, and it has left its mark in history, making differences in the development of civilizations in different geographical locations and times [1,2]. Agricultural progress has been a fundamental basis for the growth of human populations, because the production of sufficient food in quantity and quality depends on it. By the year 2050, the human population will have grown to about 10 billion people, according to United Nations data [3], which implies great challenges for different food production activities.

In the last two decades, technological advances applied to the tillage of the land have generated what is called precision agriculture (*PA*). It encompasses a set of technologies that combine sensors, statistics, classical algorithms, and algorithms of artificial intelligence (*AI*) especially those of computer vision (*CV*) [1]. The increase in the amount of food from the cultivation of the fields and the optimization of resources used for this are some of the main goals of the *PA*. These can be achieved by monitoring certain characteristics of the crops, such as growth, irrigation, fertilization, detection of pests and diseases [4–6].

Currently, several *CV* methods have been developed to support agricultural activity in tasks such as the estimation of fruit quality [7–9], recognition of pests [10–12], improvement of irrigation systems [13] and nutrient deficiency detection [14].

Color attributes in digital images are used to segment different crop elements, such as leaves, fruits, and weeds, among others, from the rest of the elements present in it [15]. Rasmussen [16] evaluated the level of leaf development in fields free of weeds, highlighting the importance of the conditions under which these images are acquired for obtaining successful results such as camera angle and light conditions among others. Kirk [17] estimated the amount of vegetation or foliage present in images of cereal crops at early stages of phenological development. In [18], Story presented a work to determine overall plant growth and health status. In this method, RGB (red, green, blue) and HSL (hue, saturation, lightness) formats values as are used as color features. Wang [19] proposed a method to segment rice plants from the background of the image based on subtracting the value of the green channel from the value of the red channel for the pixels of the image. The results of the segmentation process are used to estimate the amount of nitrogen present in the leaves. Quemada [20] carried out a segmentation process in hyperspectral images for the estimation of nitrogen present in corn crops. Additionally, there are computational methods implemented to detect weeds in uncontrolled light conditions, as was reported by Jeon [21], in which images were acquired using an autonomous robot. Yadav [22] measured the amount of chlorophyll found in potato crops using *CV* algorithms. Philipp [23] performed a series of segmentation comparisons using different color representation models. In [24], Menesatti proposed the use of a rapid, non-destructive, cost-effective technique to predict the nutritional status of orange leaves utilizing a Vis–NIR (visible–near infrared) portable spectrophotometer. Fan [25] developed a method for segmenting apples combining local image features and color information through a pixel patch segmentation method based on a gray-centered *RGB* color model space to address this issue.

At a more specific level, there are papers that describe methods for the segmentation and analysis of different elements that are part of the plants in crops, which use different *CV* techniques. For example, Xu [26] reported a method for extracting color and textures characteristics of the leaves of tomato plants, which is based on histograms and Fourier transforms. Wan [27] proposed a procedure to measure the maturity of fresh supermarket tomatoes at three different levels through the development of a threshold segmentation algorithm based on the *RGB* color model, and classification is performed using a backpropagation neural network. Tian [28] used an improved *k*-means algorithm based on the adaptive clustering number for the segmentation of tomato leaf images. Castillo-Martínez [29] reported a color index-based thresholding method for background and foreground segmentation of plant images utilizing two color indexes which are modified to provide better information about the green color of the plants. Lin [30] proposed a detection algorithm based on color, depth, and shape information for detecting spherical or cylindrical fruits on plants. Lu [31] presented a method for automatic segmentation of plants from background in color images, which consists of the unconstrained optimization of a linear combination of *RGB* color model component images to enhance the contrast between plant and background regions.

In recent years, a new image processing technique called deep learning (*DL*) has been developed. It consists of several types of convolutional neural network (*CNN*) models [32], for example: LeNet [33], AlexNet [34], VGG-16 [35], and Inception [36]. The capabilities and applications of CNN models have increased as well as the number of trainable parameters in them. These are a function of the number of layers used; therefore, highly specialized hardware has been required for their training. In PA, the use of DL has been successfully used in different contexts, for example, for pest and disease detection [37–39], leaf identification [11,40], and estimation of nutrients present in plant leaves [41]. The development of CNN models for performing the separation of items of interest from items of non-interest has been explored in several papers. In [42]. Milioto used an RCNN model to segment sugar beet plants, weeds, and background using *RGB* images. Majjed [43] trained different CNN models based on SegNet and FNC to segment grapevine cordons and determine their

trajectories. Kang [44] proposed several CNN models based on DaSNet and a ResNet50 backbone for real-time semantic apple detection and segmentation.

This paper details a segmentation method applied to images taken in greenhouses with tomato crops, classifying the pixels into three classes: leaves, fruits, and background. This is based on segmentation using the dominance of a color channel with respect to the others and, in a following stage, the determination of thresholds using the same color channel information. This method has the advantages of ease of implementation and low computational cost.

The reminder of this paper is organized as follows. The segmentation method developed to separate the leaves and fruits of the tomato plants is described in detail in Section 2. In Section 3, different images generated during the segmentation process of the leaves and fruits are shown. Additionally, tables are displayed with the metrics selected to measure performance. A comparison of results generated by the developed segmentation method against those of a CNN model is made in Section 4. In Section 5, a commentary on the performance of our method is provided. Lastly, the conclusions are presented in Section 6.

## 2. Method

The different methods of separating the elements present in images into portions that are easier to analyze are called segmentation methods [45–47]. These can be classified into the following categories:

- Region-based methods. These methods are based on separating a group of pixels that are connected and share properties. This technique performs well on noisy images.
- Edge-based methods. These algorithms are generally based on the discontinuity of the pixel intensities of the images to be segmented, which are manifested at the edges of the objects.
- Feature-based clustering methods. These methods are based on looking for similarities between the objects present in the images; this allows for the creation of categories of interest for a particular objective.
- Threshold methods. These methods are based on a comparison of the pixel intensity value against a threshold value $T$. There are two types of threshold segmentation methods depending on the value $T$; if it is constant, it is called global threshold segmentation, otherwise it is called local threshold segmentation.

The following sections detail the proposed method of using color dominance to segment all pixels of digital images acquired inside greenhouses into three different classes: leaves, tomato plant fruits, and background, which is based on the calculation of local thresholds for each pixel.

*RGB Thresholding*

The *RGB* color model is made up of three components: one for each primary color. These can have values ranging from 0 to 255, allowing any color in the visible spectrum to be represented.

The method uses a two-stage algorithm to classify each pixel of the image into one of three classes: leaves, fruits, and background. The first stage is based on the dominance of one of the color channels over the other two in the *RGB* color model; the green channel is used for the leaves, and the red channel is used for the fruits. The second stage aims to eliminate the false positives generated by the first stage. The segmentation of leaves and fruits of the tomato plant is based on the calculation of four thresholds for the differences of the dominant color channel against the other two, two to label the leaves and the remaining two for the fruits. In the calculation of the thresholds, statistical variables such as the standard deviation and maximum values of the dominant color channels are used.

An image can be mathematically represented as a two-dimensional function $f(x, y)$. If it is handled with the *RGB* color model, it is made up of three elements: $f_R(x, y), f_G(x, y)$, and $f_B(x, y)$, where the subscript refers to the primary colors red, green, and blue, respectively. $x$ and $y$ represent the spatial coordinates of a particular pixel within the image of

dimension $MxN$, where $M$ represents the number of rows and $N$ is the number of columns in an image.

The first segmentation stage is based on the dominance of the green color channel over the other two channels. This is performed by applying:

$$h(x,y) = \begin{cases} f(x,y) & if & (f_G(x,y) \geq f_R(x,y)) & and & (f_G(x,y) \geq f_B(x,y)) \\ 0 & in & another & case \end{cases}, \quad (1)$$

where $h(x,y)$ contains the pixels that are filtered from $f(x,y)$ with dominance of the green color channel over the other two for $\forall x = 0,1,2,\ldots,M$ and $\forall y = 0,1,2,\ldots,N$.

The other group of pixels of interest to segment are those from the fruits of the tomato plants. The segmentation of the fruits is based on the dominance of the red color channel over the other two; it is performed by applying:

$$j(x,y) = \begin{cases} f(x,y) & if & (f_R(x,y) \geq f_G(x,y)) & and & (f_R(x,y) \geq f_B(x,y)) \\ 0 & in & another & case \end{cases}, \quad (2)$$

where $j(x,y)$ contains the pixels that are filtered from $f(x,y)$ with dominance of the red color channel over the other two $\forall x = 0,1,2,\ldots,M$ and $\forall y = 0,1,2,\ldots,N$.

The second stage of the segmentation process begins with the calculation of the differences between the dominant color channel and the non-dominant channels in $h(x,y)$ and $j(x,y)$, which are calculated by applying:

$$\Delta_G^R(x,y) = h_G(x,y) - h_R(x,y), \quad (3)$$

$$\Delta_G^B(x,y) = h_G(x,y) - h_B(x,y), \quad (4)$$

$$\Delta_R^G(x,y) = j_R(x,y) - j_G(x,y), \quad (5)$$

$$\Delta_R^B(x,y) = j_R(x,y) - j_B(x,y), \quad (6)$$

where $\Delta_G^R(x,y)$ and $\Delta_G^B(x,y)$ are used to determine the dominance of the green color channel related to the pixels that form the leaves in $h(x,y)$. $\Delta_R^G(x,y)$ and $\Delta_R^B(x,y)$ are utilized to determine the dominance of the red color channel related to the pixels that form the fruit in $j(x,y)$. The subscript refers to the dominant color channel, whereas the superscript refers to one of the other two color channels.

Then, the thresholds are implemented to determine which pixels belong to leaves and fruits. These are calculated with:

$$U_G^R(x,y) = \frac{h_G(x,y)}{M_G} * \sigma_G^R * \alpha, \quad (7)$$

$$U_G^B(x,y) = \frac{h_G(x,y)}{M_G} * \sigma_G^B * \alpha, \quad (8)$$

$$U_R^G(x,y) = \frac{j_R(x,y)}{M_R} * \sigma_R^G * \alpha, \quad (9)$$

$$U_R^B(x,y) = \frac{j_R(x,y)}{M_R} * \sigma_R^B * \alpha, \quad (10)$$

where $U_G^R(x,y)$ and $U_G^B(x,y)$ are the thresholds used to detect leaves, and $U_R^G(x,y)$ and $U_R^B(x,y)$ are utilized to find the fruit region. $M_G$ and $M_R$ are the highest values of the green and red color channels in $h(x,y)$ and $j(x,y)$, respectively. $\sigma_G^R, \sigma_G^B, \sigma_R^G$, and $\sigma_R^B$ correspond to the standard deviations of the $\Delta_G^R, \Delta_G^B, \Delta_R^G$, and $\Delta_R^B$ values, respectively. Finally, $\alpha$ is a factor utilized to control the thresholds with the objective of maximizing the result of a particular metric by experimenting with different values of $\alpha$.

The image $h_F(x,y)$ with the pixels that make up the leaves filtered from $h(x,y)$ $\forall x = 0,1,2,\ldots,M$ and $\forall y = 0,1,2,\ldots,N$ is obtained with:

$$h_F(x,y) = \begin{cases} h(x,y) & if \quad (\Delta_G^R(x,y) > U_G^R(x,y)) \quad and \quad (\Delta_G^B(x,y)) > U_G^B(x,y))) \\ 0 & in \qquad another \qquad case \end{cases} \quad . \ (11)$$

The image $j_F(x,y)$ with the pixels that make up the fruits filtered from $j(x,y)$, $\forall x = 0, 1, 2, \ldots, M$ and $\forall y = 0, 1, 2, \ldots, N$ is obtained with:

$$j_F(x,y) = \begin{cases} j(x,y) & if \quad ((\Delta_G^R(x,y) > U_G^R(x,y)) \quad and \quad (\Delta_G^B(x,y) > U_G^B(x,y))) \\ 0 & in \qquad another \qquad case \end{cases} \quad . \ (12)$$

Figure 1 describes the segmentation process of tomato leaves and fruits in a general way.

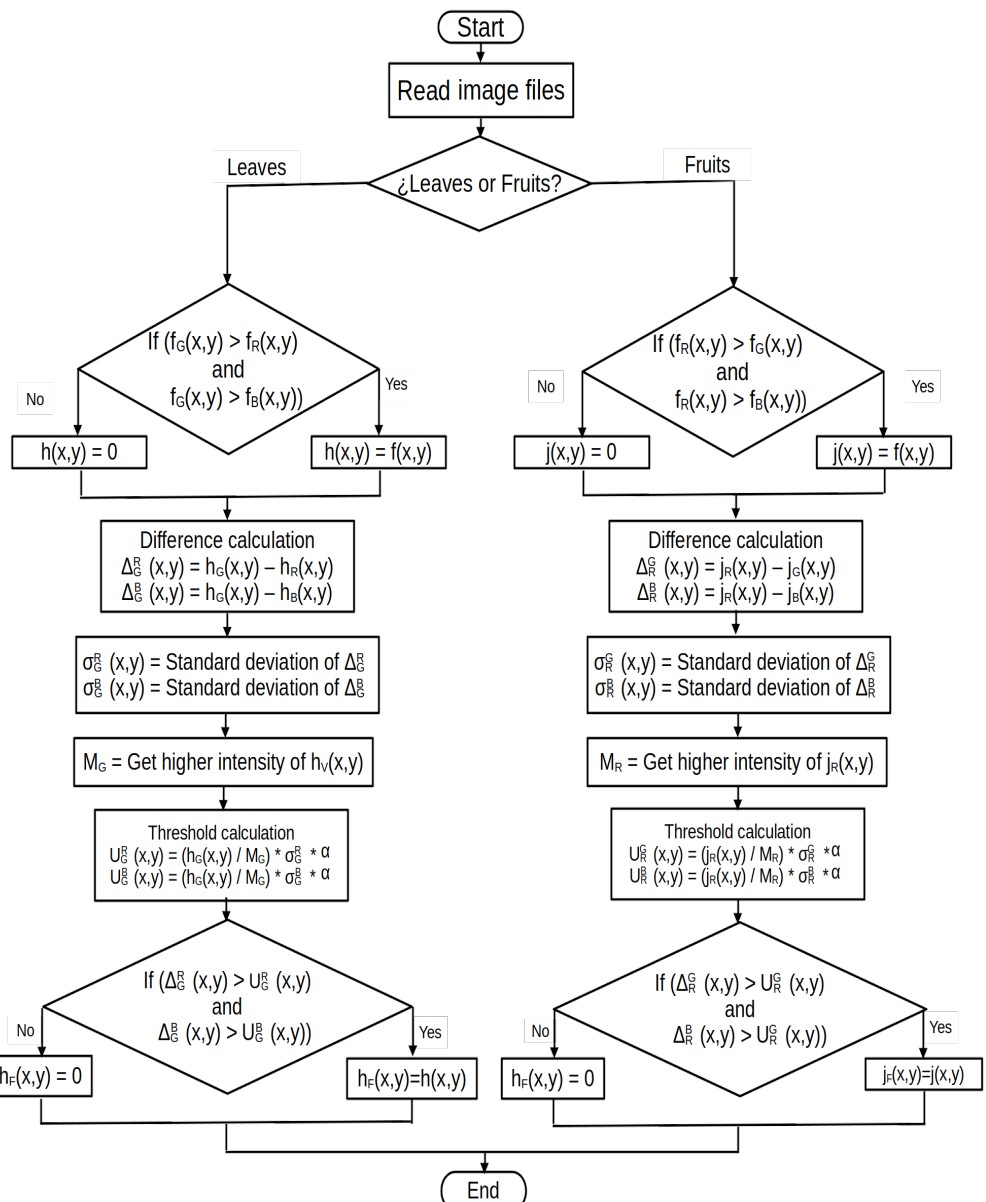

**Figure 1.** Implementation methodology of the proposed segmentation algorithm by means of thresholds.

## 3. Experimentation

The characteristics of the computer, dataset images, and manual labeling process used to create the masks necessary for the evaluation of performance metrics and presentation

of results of the segmentation of leaves and fruits of the tomato plant are described in the following sections.

### 3.1. Programming Language and Computer Characteristics

Python version 3.9.15 was the selected language for the programming of the proposed algorithm due to the number of libraries, of which OpenCV was used for image management.

A computer with the following characteristics was used for the experimental phase of the color dominance segmentation method:

- Processor: Intel® Core™ i7-8550U CPU @ 1.80 GHz × 8.
- RAM: 16 GB.
- Video card: NVIDIA® GeForce® 150MX.
- Operating system: Ubuntu 22.04.2 LTS 64 bits.

### 3.2. Dataset

The dataset named "Tomato Detection" consists of 850 images with shots of tomato plantations grown inside greenhouses, which are accessible from the web address https://www.kaggle.com/datasets/andrewmvd/tomato-detection, accessed on 13 February 2023.

In the experimental section, 100 images were used, representing a sample of 12% of the dataset. The selected images were in the PNG file format and were 500 × 400 pixels in size resolution (see Figure 2).

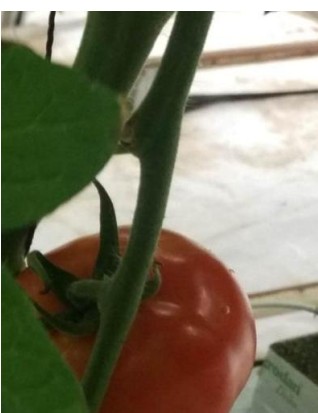

**Figure 2.** Example image from the tomato plantation dataset.

Labeling of the Dataset

The proposed color dominance segmentation method does not require any prior labeling. To measure the performance of the results generated by segmenting the pixels that form the leaves, fruits, and background of the images, the selected images are labeled.

Image labeling was performed using the Computer Vision Annotation Tool (CVAT), available at the website https://www.cvat.ai/, accessed on 28 February 2023. CVAT is a free web tool with several features that allow different types of labeling for many uses such as recognition, detection, segmentation, and others. In the case of the segmentation of the leaves and fruits of tomato plants, it is necessary to create two labels in a CVAT labeling project task, one for each item of interest, because everything that is not manually labeled is considered background. In this case, the tool *Draw new polygon* was used and it was established to label where it will belong, as well as a number of points that are established according to the difficulty of the object to be labeled. It is important to mention that the labeling process must be performed on all images used in the experimental section; this requires a large amount of time to perform. Figure 3 shows the CVAT graphical interface in the labeling process of Figure 2.

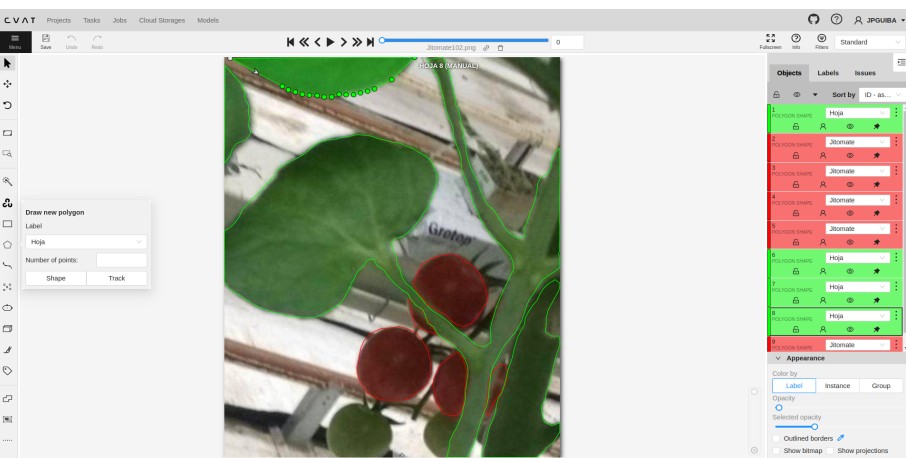

**Figure 3.** CVAT labeling interface.

Figure 4 shows the result of labeling the pixels of the leaves in green, the pixels of fruits in red, and the background in black.

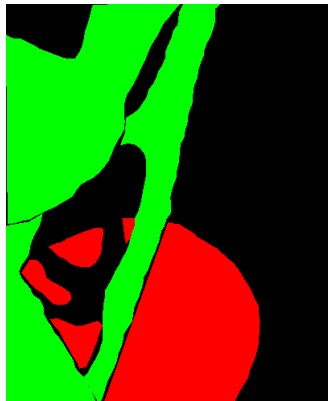

**Figure 4.** Labeling of Figure 2, with the leaves in green, fruits in red and background in black.

*3.3. Performance Metrics*

To measure the effectiveness of the proposed segmentation method *Accuracy*, *Precision*, *Recall*, *F1-Score* [48], and *Intersection over Union* [49] are used; these are metrics regularly used when measuring the performance of some method or technique in image segmentation. In a binary classification, a pixel can be labeled as either positive or negative, where positive is belonging to a particular class and negative is not belonging to it. The decision is represented in a confusion matrix, which has four elements:

1. *True positives* ($TP$) are pixels labeled as positive in the real image and in the same way by the segmented method.
2. *False positives* ($FP$) are the pixels labeled by the segmentation method as positive, when in the real image they are not.
3. *True negatives* ($TN$) are the pixels marked as negative in real image and in the same way by the segmented method.
4. *False negatives* ($FN$) are pixels incorrectly labeled as negative by the segmentation method.

The *Accuracy* metric indicates the total number of pixels correctly classified in relation to the total number of classifications made by the segmentation method. It indicates what percentage of the classifications made by the model are correct and is recommended for use in problems in which the data are balanced. The metric is expressed as:

$$Accuracy = \frac{TP + TN}{TP + TN + FP + FN}. \tag{13}$$

The *Precision* metric is the percentage of the number of true positives that are actually positive compared to the total number of predicted positive by the segmentation method. This reflects the degree of proximity of the results of different measurements to each other. The metric is expressed as:

$$Precision = \frac{TP}{TP + FP}. \tag{14}$$

The *Recall* metric represents the percentage difference between the number of true positives that the segmentation method has classified and the total number of predicted positive values. This metric is recommended when there exists a high cost associated with false negatives. The metric is expressed as:

$$Recall = \frac{TP}{TP + FN}. \tag{15}$$

The *F1-Score* metric represents the harmonic average of the precision and the recall. Its main advantage is that it summarizes both metrics in a single value. The metric is expressed as:

$$\text{F1-Score} = 2 * \frac{precision * recall}{precision + recall}. \tag{16}$$

The *IoU* metric refers to the similarity between the predicted image and the corresponding mask; it is an important metric when comparing the results of the method against manually created marks. The metric is expressed as:

$$IoU = \frac{TP}{TP + FP + FN}. \tag{17}$$

### 3.4. Region Segmentation Process for the Leaves and Fruits of the Tomato Plants

The proposed method developed to separate the leaves, fruits, and background in tomato plant crops was applied to the dataset on several occasions, with the objective of maximizing the result of the *IoU* metric. The results are generated with an $\alpha = 3.5$, which maximizes the results for the interest classes (see Figure 5).

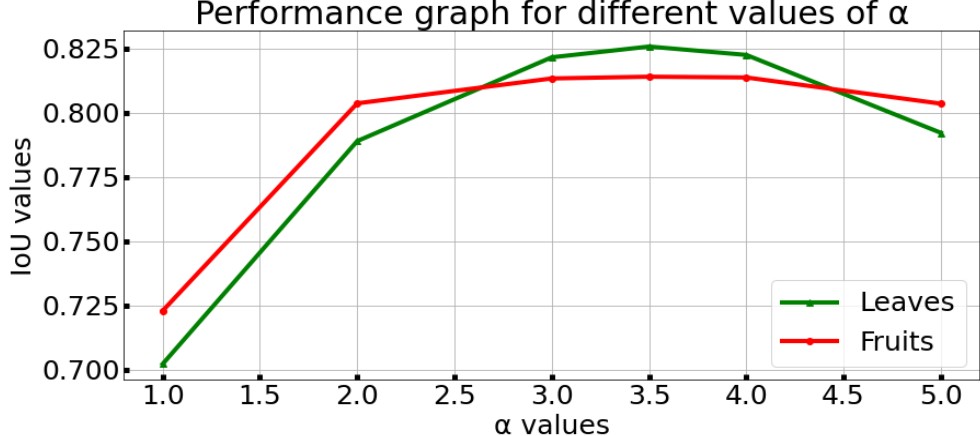

**Figure 5.** Results of calibration sensitivity test for $\alpha$.

Figure 6 shows images resulting from applying the two stages of the segmentation method to five random images from the dataset to segment the pixels that belong to leaves of the tomato plant.

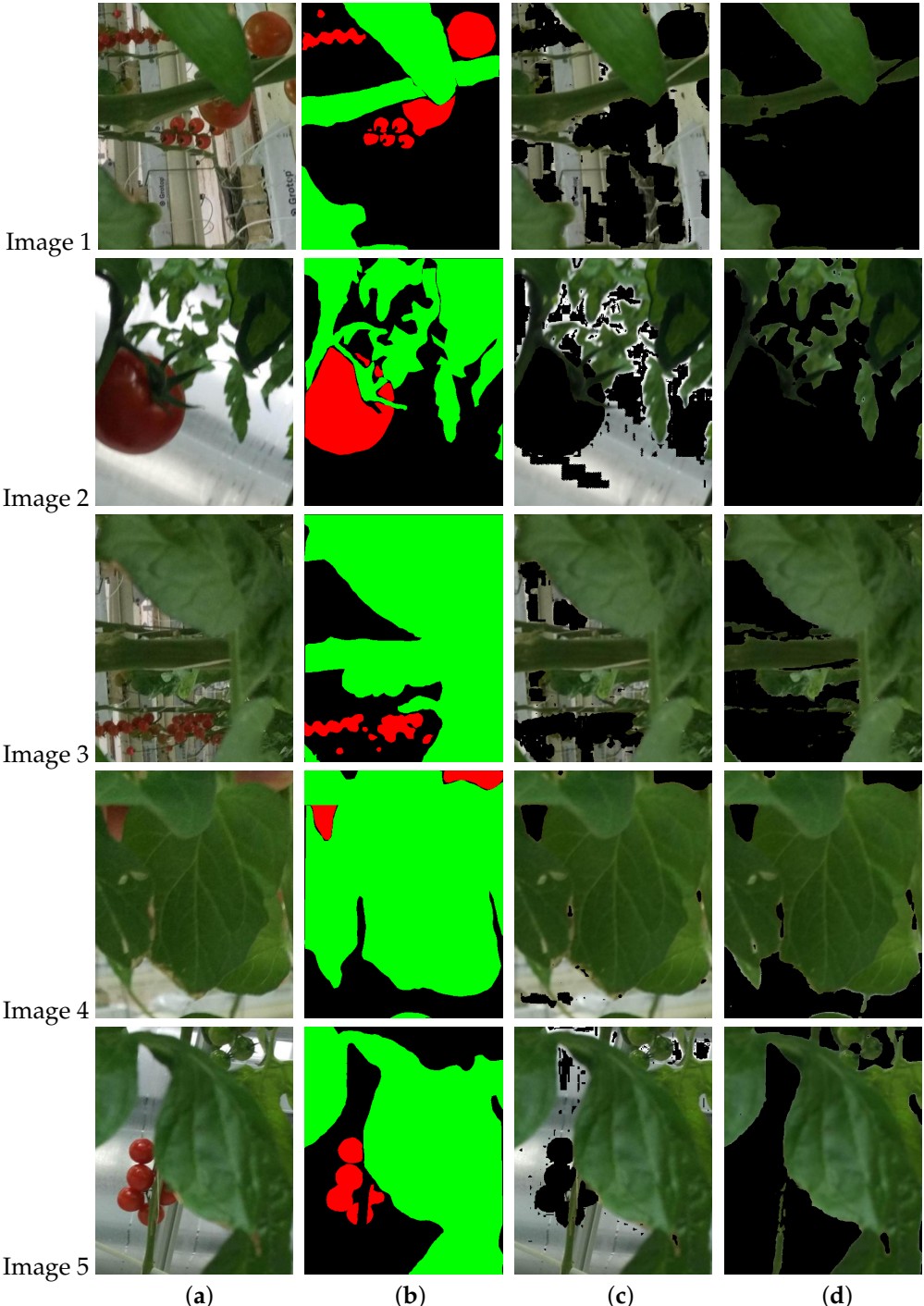

**Figure 6.** Images resulting from the segmentation process for the leaves. (**a**) Original image, (**b**) image mask, (**c**) result of applying Equation (1) to $f(x,y)$, (**d**) result of applying Equation (11) to $h(x,y)$.

Furthermore, Figure 7 shows images resulting from applying the two stages of the segmentation method to the same five images in Figure 6 to segment the pixels that belong to fruits of the tomato plant.

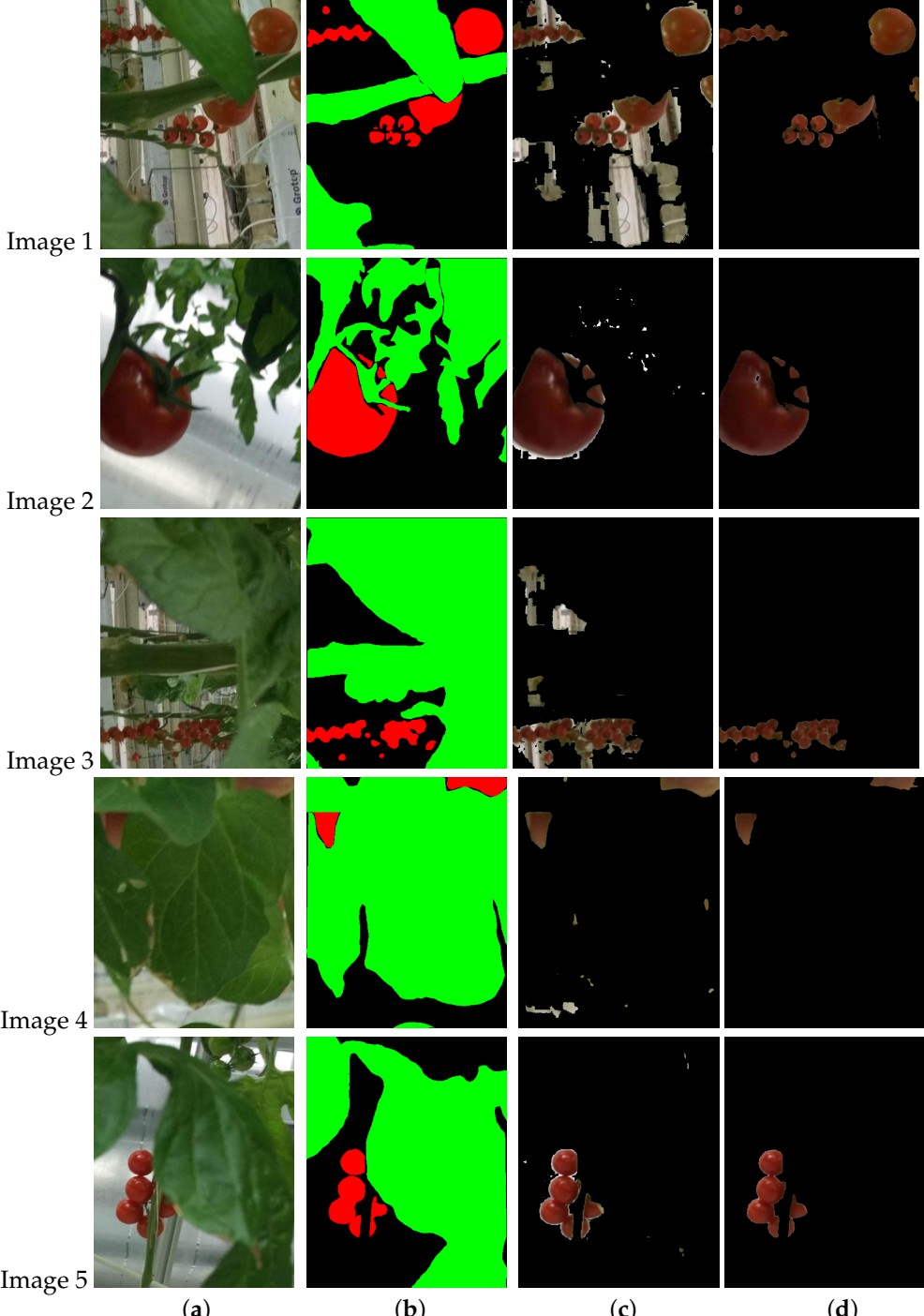

**Figure 7.** Images resulting from the segmentation process for the fruits. (**a**) Original image, (**b**) image mask (**c**), result of applying Equation (2) to $f(x, y)$, (**d**) result of applying Equation (12) to $j(x, y)$.

The results when evaluating the images of Figures 6 and 7 with the selected metrics are shown in Tables 1 and 2 for leaves, whereas for the fruits, the results are in Tables 3 and 4.

**Table 1.** Performance metrics for the segmentation of the leaves of tomato plants with the five images of Figure 6c applying Equation (1).

| | | Result Metrics of First Segmentation of Leaves | | | |
|---|---|---|---|---|---|
| **Image** | *Accuracy* | *Precision* | *Recall* | *F1-Score* | *IoU* |
| 1 | 61.90% | 41.95% | 99.98% | 59.11% | 41.95% |
| 2 | 68.03% | 54.09% | 98.81% | 69.91% | 57.74% |
| 3 | 81.92% | 79.48% | 99.99% | 88.56% | 79.48% |
| 4 | 83.43% | 82.91% | 99.71% | 90.54% | 82.71% |
| 5 | 67.15% | 64.44% | 99.89% | 78.34% | 64.39% |
| Averages | 72.48% | 64.57% | 99.67% | 77.29% | 62.25% |

**Table 2.** Performance metrics for the segmentation of the leaves of tomato plants with the five images in Figure 6d applying Equation (11).

| | | Result Metrics of Second Segmentation of Leaves | | | |
|---|---|---|---|---|---|
| **Image** | *Accuracy* | *Precision* | *Recall* | *F1-Score* | *IoU* |
| 1 | 95.96% | 93.25% | 91.99% | 92.61% | 86.24% |
| 2 | 95.11% | 97.06% | 89.70% | 93.23% | 87.33% |
| 3 | 95.19% | 95.08% | 98.21% | 96.62% | 93.46% |
| 4 | 97.20% | 97.48% | 99.03% | 98.25% | 96.56% |
| 5 | 93.52% | 91.45% | 98.30% | 94.75% | 90.02% |
| Averages | 95.39% | 94.86% | 95.44% | 95.09% | 90.72% |

The average of the five metrics in Table 1 is 75.85%, whereas for Table 2, the average is 94.30%. Comparing the averages of the application of Equation (1) with those resulting from Equation (11), a general increase of 18.44% is observed.

**Table 3.** Performance metrics for the segmentation of the fruits of tomato plants with the five images of Figure 7c applying Equation (2).

| | | Result Metrics of First Segmentation of Fruits | | | |
|---|---|---|---|---|---|
| **Image** | *Accuracy* | *Precision* | *Recall* | *F1-Score* | *IoU* |
| 1 | 77.72% | 28.92% | 99.47% | 44.82% | 28.88% |
| 2 | 97.83% | 85.15% | 99.93% | 91.95% | 85.10% |
| 3 | 92.28% | 30.95% | 99.95% | 47.26% | 30.94% |
| 4 | 98.77% | 69.53% | 99.29% | 81.79% | 69.19% |
| 5 | 98.80% | 78.08% | 99.87% | 87.64% | 78.00% |
| Averages | 93.08% | 58.52% | 99.70% | 70.69% | 58.42% |

**Table 4.** Performance metrics for the segmentation of the fruits of tomato plants with the five images of Figure 7d applying Equation (12).

| | | Result Metrics of Second Segmentation of Fruits | | | |
|---|---|---|---|---|---|
| **Image** | *Accuracy* | *Precision* | *Recall* | *F1-Score* | *IoU* |
| 1 | 97.62% | 80.54% | 97.45% | 88.19% | 78.84% |
| 2 | 98.89% | 92.26% | 99.40% | 95.68% | 91.72% |
| 3 | 98.78% | 74.26% | 99.40% | 85.01% | 72.93% |
| 4 | 99.71% | 95.24% | 94.22% | 94.72% | 89.98% |
| 5 | 99.55% | 93.05% | 96.70% | 94.84% | 90.19% |
| Averages | 98.91% | 87.07% | 97.43% | 91.68% | 84.73% |

The average of the five metrics in Tables 3 is 76.08%, whereas for Table 4, the average is 91.96%. Comparing the averages of the application of Equation (2) with those resulting from Equation (12), a general increase of 15.88% is observed.

Table 5 shows the averages of the performance metrics for the segmentation of the leaves of tomato plants in the 100 images that belong to the test set; the lowest metric is *IoU*, with 82.62%, whereas the highest is *Recall*, with 95.08%.

**Table 5.** Averages of performance metrics for the segmentation methods over the 100 images of test set.

| Averages of Performances for Leaves | | | | |
|---|---|---|---|---|
| *Accuracy* | *Precision* | *Recall* | *F1-Score* | *IoU* |
| 93.70% | 86.13% | 95.08% | 88.31% | 82.62% |

Figure 8 shows the distribution of the results of the segmentation process with the metrics for the leaves of the 100 images in the dataset. In all metrics, the mean is less than the median, which indicates that they have a distribution skewed to the left; two of them are in the first quartile, and the remaining three are in the second. Regarding the dispersion of the metric results, the greatest is in *IoU* followed by *Precision*, *F1-Score*, and *Accuracy*, and the metric with the least dispersion is *Recall*.

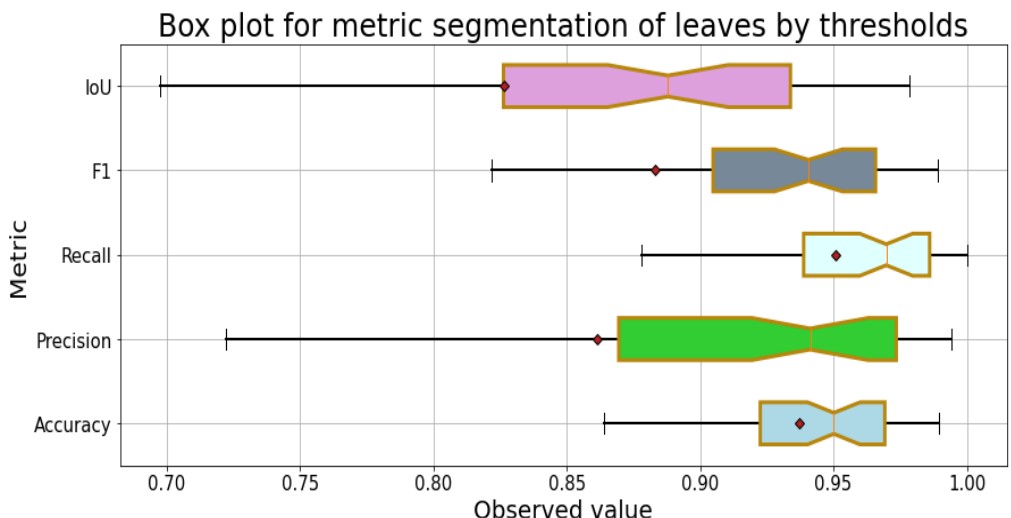

**Figure 8.** Box plot of the metrics for the segmentation of leaves.

Table 6 shows the averages of the performance metrics in the segmentation of the fruits of tomato plants to the 100 images that belong to the test set. The lowest metric is *IoU*, with 80.60%, whereas the highest is *Accuracy*, with 98.34%.

**Table 6.** Averages of performance metrics for the segmentation methods over the 100 images of the test set.

| Averages of Performance Metrics for Fruits | | | | |
|---|---|---|---|---|
| *Accuracy* | *Precision* | *Recall* | *F1-Score* | *IoU* |
| 98.34% | 84.36% | 94.51% | 88.32% | 80.60% |

Figure 9 shows the distribution of the results of the segmentation process with the metrics for the fruits of the 100 images in the dataset; in all metrics, the mean is less than the median, which indicates that they have a distribution skewed to the left, and all are in the second quartile. Regarding the dispersion of the metric results, the greatest is in *IoU*

followed by *Precision*, *F1-Score*, and *Recall*. The metric with the least dispersion is *Accuracy*.

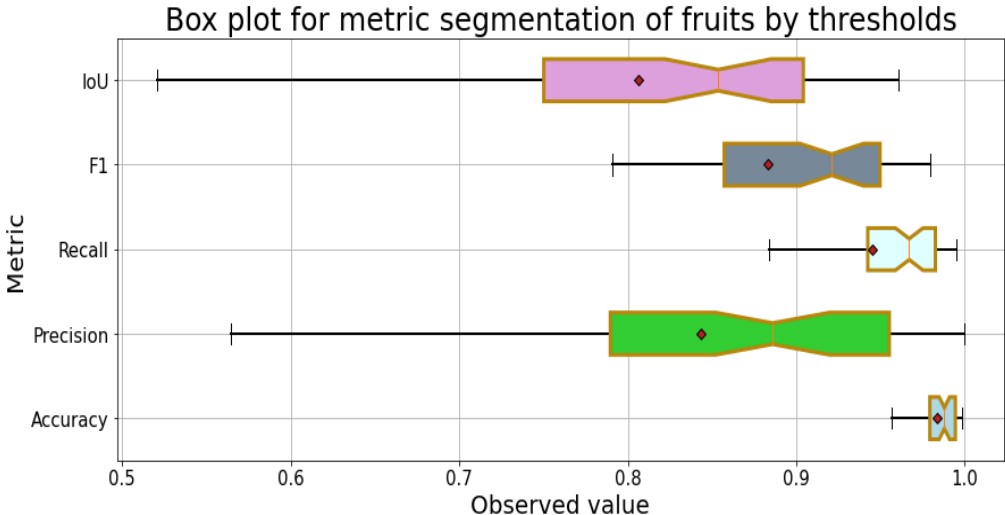

**Figure 9.** Box plot of the metrics for the segmentation for fruits.

## 4. Comparison of Results against PSPNet Model

A CNN PSPNet [50] model was trained with a ResNet50 [51] backbone to perform semantic segmentation of the leaves and fruits of tomato plants. Figure 10 shows the architecture of the implemented CNN PSPNet model.

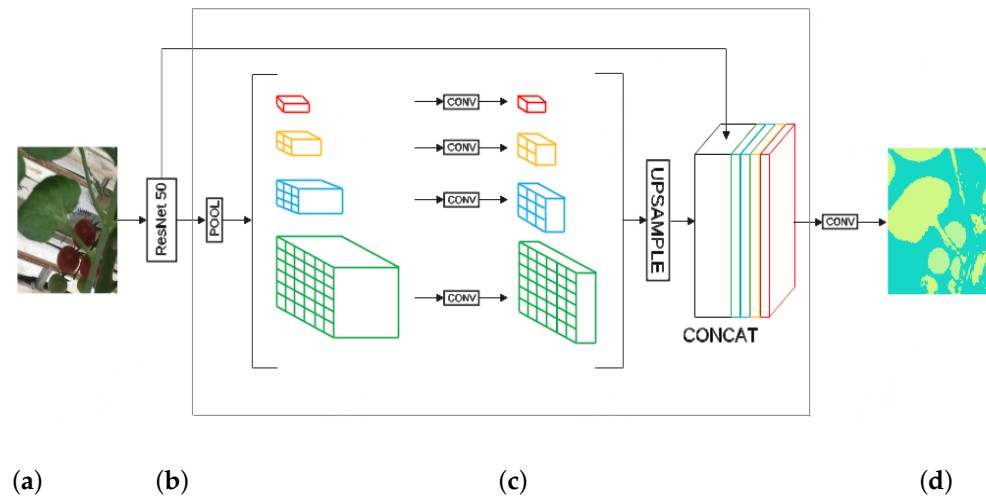

(**a**)      (**b**)          (**c**)          (**d**)

**Figure 10.** Architecture of the PSPNet model. (**a**) Input image, (**b**) backbone, (**c**) pyramid pooling module, (**d**) output image.

The CNN model was created using the *TensorFlow* and *OpenCV* libraries. For the training process, two sets of images of the initial dataset were created: the *Training* dataset with 180 items and the *Validation* dataset with 80, where it was necessary to perform the same labeling as described in the corresponding section for both sets of images. The learning process was carried out for 70 epochs with a learning rate of 0.01, lasting eight hours using the equipment described in corresponding section. The accuracy for the training set was about 0.95, whereas for the validation set it was 0.90.

Five images were processed with the color dominance segmentation method and compared with the segmentation using the CNN model. The qualitative and quantitative results are presented in Section 3.

A qualitative comparison of the segmentation on tomato leaves and fruits by the CNN model and color dominance method is shown in Figures 11 and 12. The differences in the segmentation results in both cases are observed near the contours of the leaves and fruits.

Tables 7 and 8 show the quantitative results of the five metrics used to measure the segmentation performance of leaves and fruits of tomato plants. In most cases, the performance of the color dominance segmentation method is superior to that of the CNN model.

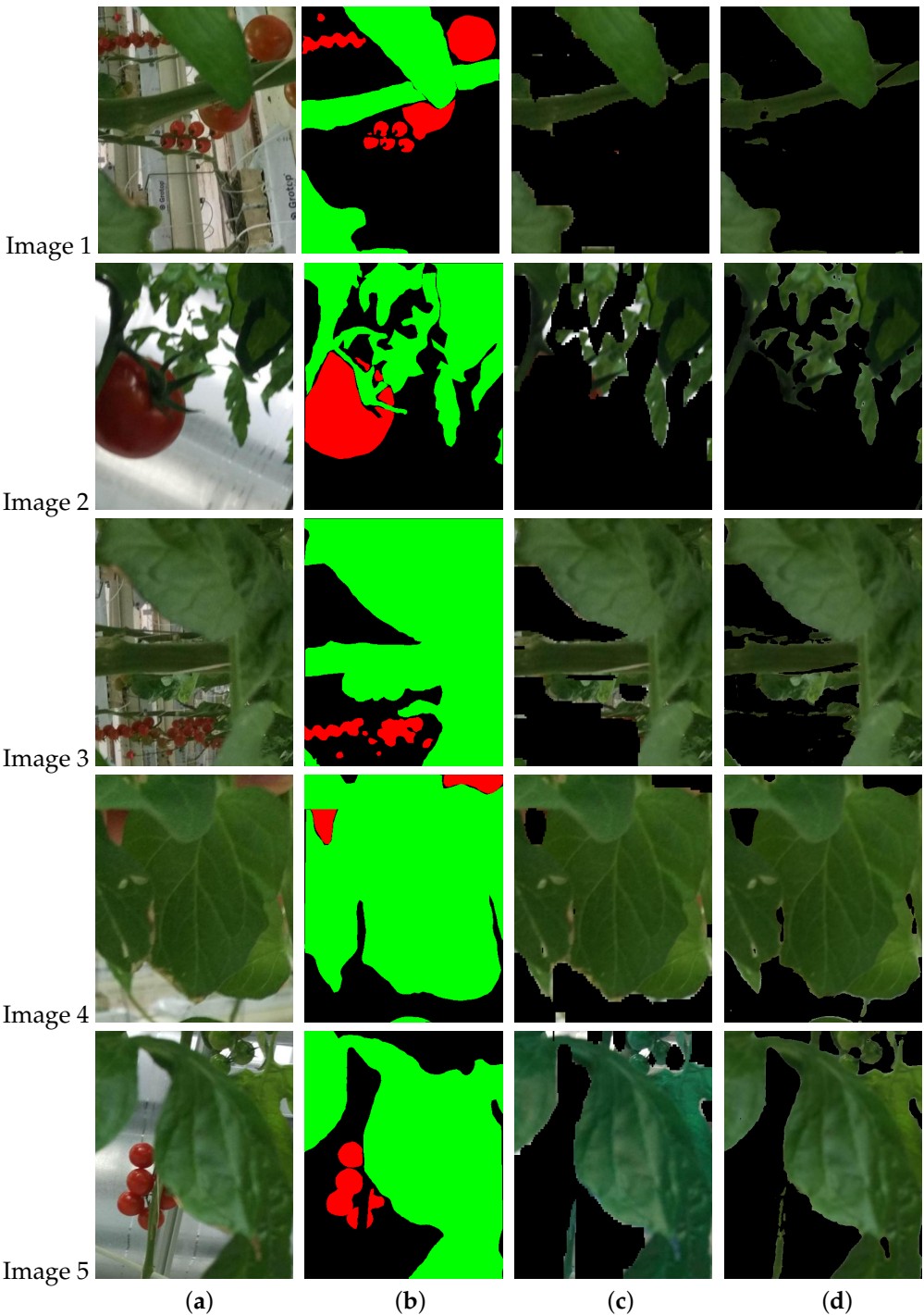

**Figure 11.** Comparison of segmentation results for the leaves. (**a**) Original image, (**b**) image mask (**c**) segmentation by CNN PSPNet, (**d**) segmentation by color dominance.

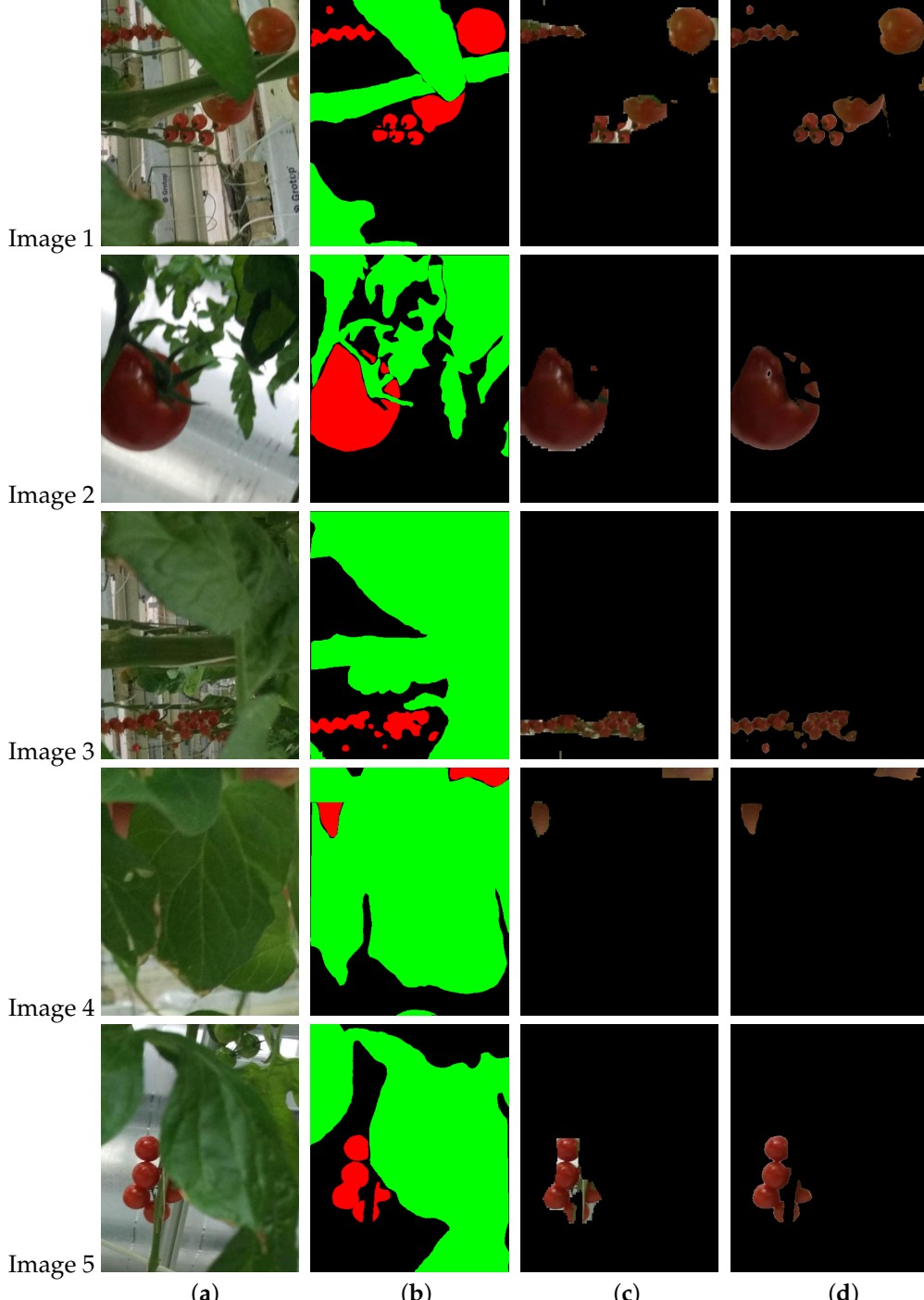

**Figure 12.** Comparison of segmentation results for the fruits. (**a**) Original image, (**b**) image mask, (**c**) segmentation by CNN PSPNet, (**d**) segmentation by color dominance.

The PSPNet model was used to perform the segmentation of 100 images in which the color dominance segmentation method was tested to obtain an overall quantitative comparison of the performance of both methods with the same dataset. The comparative averages of the performance metrics are shown in Table 9.

**Table 7.** Comparison of five leaf image segmentation results.

| | Comparison of Leaf Segmentation Results | | | | | | | | | |
|---|---|---|---|---|---|---|---|---|---|---|
| | **By Color Dominance** | | | | | **PSPNet** | | | | |
| **Image** | *Accuracy* | *Precision* | *Recall* | *F1-Score* | *IoU* | *Accuracy* | *Precision* | *Recall* | *F1-Score* | *IoU* |
| 1 | 95.96% | 93.25% | 91.99% | 92.61% | 86.24% | 95.81% | 90.88% | 94.26% | 92.54% | 86.12% |
| 2 | 95.11% | 97.06% | 89.70% | 93.23% | 87.33% | 91.12% | 90.65% | 85.17% | 87.82% | 78.29% |
| 3 | 95.19% | 95.08% | 98.21% | 96.62% | 93.46% | 93.24% | 92.41% | 98.44% | 95.33% | 91.07% |
| 4 | 97.20% | 97.48% | 99.03% | 98.25% | 96.56% | 94.84% | 94.76% | 98.98% | 96.83% | 93.85% |
| 5 | 93.52% | 91.45% | 98.30% | 94.75% | 90.02% | 88.07% | 74.22% | 93.81% | 82.87% | 70.76% |

**Table 8.** Comparison of five fruit image segmentation results.

| | Comparison of Fruit Segmentation Results | | | | | | | | | |
|---|---|---|---|---|---|---|---|---|---|---|
| | **By Color Dominance** | | | | | **PSPNet** | | | | |
| **Image** | *Accuracy* | *Precision* | *Recall* | *F1-Score* | *IoU* | *Accuracy* | *Precision* | *Recall* | *F1-Score* | *IoU* |
| 1 | 97.62% | 80.54% | 97.45% | 88.19% | 78.84% | 96.13% | 75.00% | 86.15% | 80.19% | 66.93% |
| 2 | 98.89% | 92.26% | 99.40% | 95.68% | 91.72% | 98.64% | 94.24% | 94.82% | 94.53% | 89.63% |
| 3 | 98.78% | 74.26% | 99.40% | 85.01% | 72.93% | 97.64% | 61.32% | 86.05% | 97.61% | 55.78% |
| 4 | 99.71% | 95.24% | 94.22% | 94.72% | 89.98% | 99.27% | 93.86% | 78.87% | 85.71% | 75.00% |
| 5 | 99.55% | 93.05% | 96.70% | 94.84% | 90.19% | 98.52% | 93.23% | 96.93% | 95.04% | 90.56% |

**Table 9.** Comparisons of leaf and fruit segmentation averages in tomato plants.

| | Comparisons of Leaf and Fruit Segmentation Averages in Tomato Plants | | | | | | | | | |
|---|---|---|---|---|---|---|---|---|---|---|
| | **Leaves** | | | | | **Fruits** | | | | |
| | *Accuracy* | *Precision* | *Recall* | *F1-Score* | *IoU* | *Accuracy* | *Precision* | *Recall* | *F1-Score* | *IoU* |
| Color dominance | 93.70% | 86.13% | 95.08% | 88.31% | 82.62% | 98.34% | 84.36% | 94.51% | 88.32% | 80.60% |
| PSPNet | 91.46% | 85.09% | 93.19% | 87.59% | 80.19% | 95.95% | 79.21% | 82.84% | 80.13% | 68.56% |

The averages of the color dominance segmentation method are higher than those of the CNN PSPNet model in all cases, both for leaves and fruits. The metric where the greatest difference is shown is $IoU$ in both cases.

Calculating the average of the results of the five metrics, the color dominance segmentation method has a superior performance of 1.66 percentage points, whereas for fruits it has a performance advantage of 7.48 percentage points when compared to the CNN model.

## 5. Discussion

The images resulting from applying the two-stage segmentation method are shown in Figures 6 and 7, which show a successful segmentation of the leaves and fruits of tomato plants.

As for the quantitative measurement, the results using the selected performance metrics from Tables 5 and 6 show adequate performance of the color dominance segmentation method. Another aspect to highlight is that the processed images were taken in real growing environments without lighting control.

When comparing the results of the color dominance segmentation method with the semantic segmentation performed with the CNN PSPNet model in Table 9, the performance of the proposed method is superior, with the great advantage of not requiring a manual image labeling process nor a prior training process costly in time and computational power.

The performance of the color dominance segmentation method can be increased by adjusting the value of $\alpha$ to maximize results or by looking for a particular segmentation objective. For example, value adjustment can be performed by applying heuristic methods, such as simulated annealing or genetic algorithms. Another alternative is to use different

values for $\alpha$, one for the leaves and a different one for the fruits, which allows for an improvement in the results of the segmentation of the leaves and fruits of the tomato plant.

Challenges for the proposed color dominance segmentation method include testing the method on images of tomato crops in the field (outside greenhouses), where brown elements such as soil can affect segmentation performance, particularly in fruit segmentation; adaptation and testing on crops with similar colors, such as strawberries and raspberries; and utilizing other color dominances that occur naturally in other crops and make the necessary adaptations to take advantage of them when segmenting the elements of interest.

An aspect of research involves looking for a color dominance segmentation process that can be used in a different color model, such as *HSV* in addition to the *RGB*, which allows the establishment of other conditions based on the dominance of one of the characteristics of the color model used.

A clear disadvantage of the proposed method is that it cannot be applied to crops in which the fruits are green because they would be classified directly as leaves, for example, the cucumber. In these cases, it will be necessary to add a method or algorithm that allows for discrimination of the shape of the fruits from the leaves.

The results of the segmentation performed by the method facilitate the search for pests, diseases, and nutritional deficiencies that may manifest themselves in the leaves, fruits, and background of the segmented images. As part of the future work using the segmentation method implemented, it is intended to develop a system that supports a diagnosis of the state of crops using intelligent algorithms such as CNN, pattern recognition, and heuristic methods for the generation of plant-saving fertilization alternatives.

In addition, it is possible to measure the results of the segmentation performed by the presented algorithm more accurately, thus reducing manual labeling errors.

## 6. Conclusions

Image segmentation methods are a basic branch in the CV research area; these can be tested in a wide variety of contexts, including agriculture. The segmentation method presented in this paper proposes a color dominance method that takes advantage of the naturally occurring differences in shades between leaves and fruits of some crops. Among its features are its simple operation, fast response, and the fact that it does not require preprocessing of input data, special hardware such as video cards, or prior training to operate.

The method of segmentation by color dominance is based on the use of statistical variables such as standard deviation and the maximum value of the dominant channel to generate local thresholds. These are then used to perform a classification of all pixels that have a dominance of the green color channel for leaves and a dominance of the red color channel for fruits.

**Author Contributions:** J.P.G.I. and F.J.C.d.l.R. also participated in the conceptualization, methodology, validation and analysis of results. O.A.A. participated along with the other authors in the mathematical foundation. The development of the source code for the tests was done by J.P.G.I. All authors have read and agreed to the published version of the manuscript.

**Funding:** This research received no external funding.

**Data Availability Statement:** Not applicable.

**Acknowledgments:** The authors wanted to thank Consejo Nacional de Humanidades Ciencia y Tecnologías (CONAHCYT), Centro de Investigaciones en Óptica A.C., Instituto Tecnológico de Estudios Superiores de Zamora, Opus Farms for giving us the opportunity to develop this research. To the teachers Luz Basurto from the Instituto Tecnologico de Estudios Superiores de Zamora for her contributions in the agricultural area. Mario Alberto Ruiz and Karla Maria Noriega from Investigaciones en Óptica A.C. for their support in the writing of the document.

**Conflicts of Interest:** The authors declare no conflict of interest.

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
