# Peer review of "Segmentation of Leaves and Fruits of Tomato Plants by Color Dominance"

_agriengineering, doi:10.3390/agriengineering5040113_

Round 1

Reviewer 1 Report

Advantages:

1. The introduction provides a clear overview of the significance of agriculture in the development of civilizations, the progress of modern agricultural technology, and the concept of precision agriculture, thereby establishing the background of the paper for readers.

2. Addressing the image segmentation issue within precision agriculture, the paper proposes a two-stage image segmentation method based on the dominance of different color channels.

3. The methodology is presented coherently, outlining the process of the two stages of image segmentation, utilizing the RGB color model, and explaining the rationale behind the choice of distinct color channels for leaf and fruit segmentation.

4. The paper references performance evaluation metrics (Accuracy, Precision, Recall, F1-Score, and Intersection over Union) and presents segmentation results for both fruit and leaf images, achieving remarkable metrics with 98.34% Accuracy for fruits and 95.08% Recall for leaves.

Disadvantages:

1. The paper lacks detailed elaboration on the specific details of the image segmentation algorithms employed, potentially necessitating further understanding of algorithm principles and methodologies by readers.

2. No mention is made of comparing the proposed method with other techniques or existing technologies to validate its superiority.

3. The paper does not discuss potential challenges or limitations that may arise in practical applications, nor does it provide solutions to address these issues.

4. Although the results section introduces several performance evaluation metrics, it does not delve into the interpretation of these metrics' significance, potentially requiring readers to possess additional background knowledge for comprehension.

In conclusion, while the paper excels in introducing the background and describing the problem, there is room for improvement in providing detailed method explanations, presenting comparative results against existing methods, and discussing challenges and application limitations

Author Response

Thank you for your comments, we try to serve you as best as possible with our conditions to improve the document.

Thank you in advance

Juan Pablo Guerra

Reviewer 2 Report

The article proposes a method for segmenting images of tomato fruits. To improve the quality of the article, the following changes should be made:

1.) In section 1. it is necessary to consider in more detail all existing methods and methods of segmentation, including the use of neural networks.

2.) What is the scientific novelty in the conducted research?

3.) The conclusions should describe in more detail the advantages of the proposed method over the known ones discussed in Section 1.

4.) What computing power is used, it is necessary to provide technical characteristics.

5.) Section 3.1.1 Labeling a dataset should describe the labeling process in more detail.

6.) What is the sample size used in the research?

7.) There is no need to add the well-known Figure 4.

Author Response

Thank you for your comments, we try to serve you as best as possible with our conditions to improve the document.

Thank you in advance.

Juan Pablo Guerra
